# Gender differences in health-related quality of life in people with severe mental illness

**Ester Colillas-Malet**[1]*, **Gemma Prat**[2], **Albert Espelt**[1,3,4], **Dolors Juvinyà**[5]

**1** Facultat de Ciències de la Salut de Manresa, Universitat de Vic–Universitat Central de Catalunya (UVic-UCC), Av. Universitària, Spain, **2** Grup SaMIS (Salut Mental i Innovació Social), Divisió de Salut Mental de la Fundació Althaia, Manresa, Spain, **3** CIBER de Epidemiología y Salud Pública, Instituto de Salud Carlos III, C/ Melchor Fernández Almagro 3–5, Spain, **4** Departament de Psicobiologia i Metodologia en Ciències de la Salut, Universitat Autònoma de Barcelona (UAB), Cerdanyola del Vallès, Spain, **5** Grup de recerca de salut i atenció sanitària de la Universitat de Girona, Girona, Spain

* ecolillas@umanresa.cat

**Data Availability Statement:** All relevant data are within the manuscript.

**Funding:** The authors received no specific funding for this work.

## Abstract

### Introduction and purpose

The purpose was to analyze socioeconomic and clinical factors of psychosocial functioning and self-perception in relation to health-related quality of life (HRQOL) in people with severe mental health illness (SMI) by gender.

### Materials and method

A cross-sectional study was conducted on a sample of 133 women and 90 men. Recorded variables: HRQOL, SF-36 Physical Component Scores (PCS) and Mental Component Scores (MCS); sociodemographic and clinical data on psychosocial and self-perception functioning. Correlational studies using raw and adjusted linear regression models to evaluate the factors associated with HRQOL by obtaining coefficients, p-values and respective confidence intervals.

### Results

The mean PCS for women and men was 44.6 and 49.0 (p = 0.004) and 36.4 and 37.5 (p = 0.575), respectively for MCS. The factors associated with PCS in women were age, -0.2 (-0.4:0); in work, 4.2(0.3:8.2); with an income higher than 700 euros/month, 4.4(1:7.7). In men, these factors were education level, 6.1(0.4:11.7); belief that they would not need help in the future, 4.6(0.1:9.2) and a higher need for psychosocial services, -6.6(-11.1:-2). Factors associated with MCS in women were, in work, 6.1(1.5:10.7); and having a high number of friends, 6.6(2.1:11.1). In men, these factors were, living alone, -7.1(-12.7:-1.4); lack of economic benefits, 8.5(3.2:13.8); and a higher need for psychosocial and social services, -3.6(-7.1:-0.2) and -7.7(-13.4:-2).

**Competing interests:** The authors have declared that no competing interests exist.

## Conclusions

The dimensions affected and the factors that are associated with HRQOL for people with SMI differ by gender. Therefore, these differences should to be taken into account when designing interventions for improving HRQOL.

## Introduction

One of the main objectives of the work carried out by national[1] and international[2] health organizations is the improvement of health-related quality of life (HRQOL). HRQOL is defined as the level of well-being based on an individual's evaluation of how various aspects of their life (physical, psychological, social and spiritual well-being) impact their health status[3]. There are many people with mental health problems in the community and their HRQOL is especially important to consider[4]. They often live in more disadvantaged conditions compared to the rest of the general population[5]. These conditions can lead to economic, social and emotional problems. In some cases, these people show dissatisfaction with their social relationships, economic conditions and personal safety[6,7]. The poorest aspects of HRQOL for people with severe mental illness (SMI) are social activity followed by role-emotional limitations and mental health[8].

Women present worse HRQOL than men in the general population[9,10] as well as among people with SMI. At the same time, individuals with SMI present lower HRQOL scores than the general population, as seen in a study that showed persons with obsessive-compulsive disorder and schizophrenia having lower HRQOL scores than the Spanish population, especially in the areas of mental health[8]. This gender relationship is because this factor interacts in all of the social determinants of health as a structural determinant[11] and, at the same time, as an axis of inequality[12]. In fact, it has been seen that gender role conflicts, overall work load and unpaid work have adverse effects on women's quality of life[13].

Despite a lack of consensus on the influence of socioeconomic factors and HRQOL on people with SMI[14], it has been observed that factors associated with better HRQOL are being male[15–17], young[18–20], working[21–23], having higher income[23,24], living with the family[25] and having a social network (size[26] and satisfaction[22]). Comorbidity[27,28], as well as factors such as symptoms of anxiety and depression[22,29,30] and their severity, are negatively associated with HRQOL[23]. Self-referenced psychosocial needs are also negatively associated with HRQOL in terms of number and typology[22,31–33].

Given that there is no data showing how factors differentially affect the HRQOL of women and men with SMI, the purpose of this study is to analyze, from a gender perspective, which sociodemographic, clinical and psychosocial functioning factors, just as self-perception of the current and future effects of the illness factors are associated with the health-related quality of life of people with severe mental illness.

## Materials and methods

### Design and study population

**Cross-sectional study.** The study population was composed of people diagnosed with SMI undergoing ambulatory monitoring in 2009 at the mental health center of the Mental Health Division of the Fundació Althaia in Manresa, an organization serving a population of 215,000.

## Sample selection

There was a total of 783 people who met the inclusion criteria. The inclusion criteria were: between 18 and 65 years old; having received ambulatory treatment at the mental health center; complied with one of the diagnostic criterion of the International Statistical Classifications of Diseases and Related Health Problems, 10a. Revision (ICD-10), for the diagnosis of SMI (schizophrenia and other psychotic illnesses, affective disorders such as major recurrent depression and bipolar disorders I and II, anxiety disorders such as obsessive-compulsive disorder with or without agoraphobia, and finally, borderline personality disorder); and willingness to take part in the study. The sample was selected by simple random sampling and the size was determined based on the known size of the population that met inclusion criteria, with a 5% error and a 3% accuracy. The theoretical sample was 258 people diagnosed with SMI who carried out ambulatory follow-up at mental health center. The final participation was 86.4% (223 people), 133 women (59.6%) and 90 men (40.4%). All the participants signed an informed consent form at the beginning of the assessment and the study was approved by the clinical research ethics committee of the Unió Catalana d'Hospitals.

The participants were recruited by telephone, which also served to specify the date and time for the interview. The interview consisted of a 30-minute evaluation where a trained interviewer filled out the various questionnaires.

## Variables

**Dependent variables.**   Two measurements associated with HRQOL were taken, the Physical Component Score (PCS) and the Mental Component Score (MCS). These two variables were calculated using the Spanish version of the Medical Outcome Study Short Form 36 *(SF-36)* [34] called *Cuestionario de Salud SF-36* [35]. It is a questionnaire consisting of 36 questions that measure subjective quality of life by addressing aspects of the person's daily life and identifying both the positive and negative states of PCS and MCS. These two components are divided into 8 dimensions: physical functioning, social functioning, physical role limitations, mental health role limitations, vitality, bodily pain and perceptions of general health. The scores range from 1 to 100 for both components, with 100 being a score that indicates optimal health and 0 showing a poor state of health. The mean of the general population is 50 (SD = 10) and higher or lower values are interpreted as better or worse, respectively, with respect to the reference population.

**Independent variables.**   Sociodemographic, clinical, psychosocial functioning and self-perception variables were measured. We administered our own questionnaire relative to sociodemographic and clinical variables. Information was collected on: gender, age, civil status, number of friends, work status, level of education, living arrangement, socioeconomic status, clinical diagnosis, (axis I and III of ICD-10), comorbidity, etc. An evaluation questionnaire regarding psychosocial functioning of psychosocial needs was also administered. Specifically, this was the Spanish version of the Camberwell Assessment of Need (CAN-R)[36] called *Cuestionario Camberwell para la evaluación de necesidades*[37], which contains 88 items divided into 22 areas or psychosocial needs: accommodation; food; looking after the home; self-care; daily activities; physical health; psychotic symptoms; information on condition and treatment; psychological distress; safety to self; safety to others; alcohol, drugs; company; intimate relationships; sexual expression; care of offspring; primary education; telephone; transport; money; and economic benefits. All the areas are scored from 0 to 1 with: 0, no problem; 1, moderate or serious problem due to help given. The scoring of the 22 areas is grouped into 5 dimensions: Basic Needs (3 items); Health (7 items); Social (3 items); Functioning (5 items); and Services (4 items) as found in other studies [38,39]. Finally, we administered our own

questionnaire to collect information on current and future effects of the participant's illness, as for example, if the SMI had had consequences on their social life and leisure activities, if these social consequences had improved or worsened their social relationships, if they had considered what possibilities would be in the future and if they thought that would need help.

### Analysis of the data

All data were analyzed separately for women and men[40]. A univariate descriptive analysis was performed of the variables for the purposes of describing the sociodemographic, clinical, psychosocial, current and future self-perception profiles of people with SMI included in the sample. A frequencies and percentages analyses were performed to show the distribution of the qualitative variables. A study of the means and measurements study of central tendency was undertaken for the quantitative variables. To examine the different patterns between women and men, the Chi-squared ($X^2$) test was done for the qualitative variables and the Student's T Test for the quantitative variables. To ascertain the different scoring of PCS and MCS of HRQOL, the mean scoring of each variable (PCS and MCS) was calculated with their confidence intervals at 95% for each independent variable. The variables that presented a $p < 0.1$ in the bivariate model were included in the multivariate models. In addition, the colineality of these variables was studied. Finally, to know which variables were associated with PCS and MCS, raw and adjusted linear regression models were estimated by obtaining the coefficients for the respective p-values and confidence intervals. The normality of the linear regression model residuals was checked, and they followed a normal distribution ($p > 0.01$).

All the statistical analyses were performed with the Stata 15 statistical software package.

### Results

No statistically significant differences by gender were observed in the distribution of characteristics of the sample except: civil status (48.9% women were married as opposed to 35.6% of men, p = 0.05); age ($\bar{x}_{women}$ = 44.9 years and $\bar{x}_{men}$ = 41.3 years; p = 0.01); number of children ($\bar{x}_{women}$ = 1.4 and $\bar{x}_{men}$ = 0.7; p = 0.00); and self-perceived psychosocial services needs ($\bar{x}_{women}$ = 0.3 and $\bar{x}_{men}$ = 0.1; p = 0.02). In addition, approximately 30% of the participants were in work. The most common clinical diagnosis among women was depressive disorder and bipolar disorder I and II (34.6% for both). The most common among men was schizophrenia and bipolar disorder (30.0% and 28.9%, respectively) in addition to 60% presenting physical comorbidity. Between 40% and 50% perceived a worsening of social relations and leisure activities due to SMI (Table 1).

The PCS for HRQOL shows lower scores in women with SMI than men with SMI ($\bar{x}_{women}$ = 44.6 and $\bar{x}_{men}$ = 49.0; p = 0.00) (Table 1). The women presenting better PCS were: unmarried, $\bar{x}$ = 46.3; had higher levels of education/college educated, $\bar{x}$ = 51.7; in work, $\bar{x}$ = 52.3; had bipolar disorder, $\bar{x}$ = 49.4; without physical comorbidity, $\bar{x}$ = 49.9; and had seen their social relationships and leisure activities as having improved or not changed, $\bar{x}$ = 49.0. However, age (r = -0.3) and had a lower number of psychosocial needs (r = -0.5) was negatively associated with PCS in women. On the other hand, men, like women, who presented better PCS were: in work, $\bar{x}$ = 52.2; without physical comorbidity, $\bar{x}$ = 51.6; and had seen their social relationships and leisure activities as having improved or not changed, $\bar{x}$ = 52.4. However, age (r = -0.2) and had a lower number of psychosocial needs (r = -0.5) also was negatively associated with PCS in men (Table 2).

The variables associated with PCS in women, explaining 41.4% of the variance, were: age [$\beta_a$ = -0.2 (95% CI: -0.4;0)]; in work [$\beta_a$ = 4.2 (95% CI: 0.3;8.2)]; had income of greater than 700 euros/month [$\beta_a$ = 4.4 (95% CI: 1;7.7)]; had a high number of health-related needs [$\beta_a$ =

**Table 1. Characteristics by gender.**

| | | WOMEN | | MEN | | sig. |
|---|---|---|---|---|---|---|
| | | **n** | **%** | **n** | **%** | |
| **Civil Status** | Married | 65 | 48.9 | 32 | 35.6 | 0.05 |
| | Not married (single, separated, widow) | 68 | 51.1 | 58 | 64.4 | |
| **Education level** | Primary education | 72 | 54.1 | 37 | 41.1 | 0.07 |
| | Secondary education | 43 | 32.3 | 43 | 47.8 | |
| | University studies | 18 | 13.5 | 10 | 11.1 | |
| **Work Status** | Employed | 35 | 26.3 | 27 | 30.0 | 0.55 |
| | Unemployed (+students and paid sick leave) | 98 | 73.7 | 63 | 70.0 | |
| **Living situation** | Alone | 18 | 13.5 | 18 | 20.0 | 0.20 |
| | With someone | 115 | 86.5 | 72 | 80.0 | |
| **Number of friends** | 0–3 | 51 | 38.4 | 29 | 32.2 | 0.62 |
| | 4–6 | 37 | 27.8 | 29 | 32.2 | |
| | ≥7 | 45 | 33.8 | 32 | 35.6 | |
| **Legal status** | Not disabled | 92 | 69.2 | 69 | 76.7 | 0.22 |
| | Disabled | 41 | 30.8 | 21 | 23.3 | |
| **Degree of impairment** | <33% | 94 | 70.7 | 59 | 65.6 | 0.42 |
| | >33% | 39 | 29.3 | 31 | 34.4 | |
| **Degree of disability** | Yes | 60 | 45.1 | 35 | 38.9 | 0.36 |
| | No | 73 | 54.9 | 55 | 61.1 | |
| **Economic benefits** | Receives benefits | 96 | 72.2 | 66 | 73.3 | 0.85 |
| | Does not receive benefits | 37 | 27.8 | 24 | 26.7 | |
| **Monthly income** | <700 | 49 | 36.8 | 31 | 34.4 | 0.71 |
| | >700 | 84 | 63.2 | 59 | 65.6 | |
| **Clinical diagnosis (Axis I and II)[a]** | Schizophrenia and other psychotic disorders | 25 | 18.8 | 27 | 30.0 | 0.07 |
| | Depression disorder | 46 | 34.6 | 21 | 23.3 | |
| | Bipolar disorder I and II | 46 | 34.6 | 26 | 28.9 | |
| | Others | 16 | 12.0 | 16 | 17.8 | |
| **Comorbidity (Axis III)[a]** | No Comorbidity | 42 | 31.6 | 34 | 37.8 | 0.34 |
| | Comorbidity | 91 | 68.4 | 56 | 62.2 | |
| **Social consequences and leisure activities** | Yes | 100 | 75.2 | 60 | 66.7 | 0.17 |
| | No | 33 | 24.8 | 30 | 33.3 | |
| **Social relationships[b]** | Improved or no change | 53 | 39.9 | 41 | 45.6 | 0.28 |
| | Worse | 72 | 54.1 | 40 | 44.4 | |
| | DK/NA | 8 | 6.0 | 9 | 10.0 | |
| **Possibilities for the future** | Yes | 62 | 46.6 | 48 | 53.3 | 0.33 |
| | No | 71 | 53.4 | 42 | 46.7 | |
| **Help in the future** | Yes | 72 | 54.1 | 47 | 52.2 | 0.78 |
| | No | 28 | 21.1 | 17 | 18.9 | |
| | DK/NA | 33 | 24.8 | 26 | 28.9 | |
| | | **M** | **SD** | **M** | **SD** | **sig.** |
| **Age** | | 44.9 | 9.8 | 41.3 | 10.7 | 0.01 |
| **Num. of children** | | 1.4 | 1.3 | 0.7 | 0.9 | 0.00 |
| **Num. of needs CAN[c]** | | 3.0 | 1.9 | 2.6 | 1.6 | 0.20 |
| **Type of needs[c]** | Basic | 0.0 | 0.2 | 0.0 | 0.0 | 0.06 |
| | Social | 0.4 | 0.7 | 0.5 | 0.7 | 0.59 |
| | Functioning | 0.3 | 0.6 | 0.3 | 0.6 | 0.91 |
| | Health | 1.9 | 1.0 | 1.7 | 0.9 | 0.11 |

*(Continued)*

**Table 1.** (Continued)

|  |  | WOMEN | | MEN | | sig. |
|---|---|---|---|---|---|---|
|  |  | **n** | **%** | **n** | **%** |  |
|  | Services | 0.3 | 0.5 | 0.1 | 0.4 | 0.02 |
| **SF-36[d]** | Physical Component Score | 44.6 | 11.5 | 49.0 | 9.8 | 0.00 |
|  | Mental Component Score | 36.4 | 13.9 | 37.5 | 13.5 | 0.58 |

M: Mean; SD: Standard Deviation; DK: Does not know; NA: No answer

[a.] According to Manual DSM-IV

[b.] Social relationships and activities perceptions

[c.] According the Camberwell Assessment of Need Questionnaire (CAN-R)

[d.] Quality of Life Questionnaire SF-36

-4.4 (95% CI: -6.2;-2.6)]; and did not have comorbidity [$\beta_a$ = 4.5 (95% CI: 1.1;7.9)]. These last two are also associated with men's PCS in addition to having a higher level of education [$\beta_a$ = 6.1 (95% CI: 0.4; 11.7)]; belief that they would not need assistance in the future [$\beta_a$ = 4.6 (95% CI: 0.1;9.2)]; and having a higher number of needs in the area of services [$\beta_a$ = -6.6 (95% CI: -11.1;-2.0)]. All of these explain 38.6% of the variance in men. Clinical diagnosis is not associated with PCS in either women or men (Table 3).

Men and women had similar MCS scores for HRQOL ($\bar{x}_{women}$ = 36.4 and $\bar{x}_{men}$ = 37.5) (Table 1). The women who had better MCS were those who: had secondary education, $\bar{x}$ = 40.9; were in work, $\bar{x}$ = 44.9; had more than 7 friends, $\bar{x}$ = 41.4; had bipolar disorder, $\bar{x}$ = 41.1; and those that perceived that their social relationships and leisure activities had improved or not changed, $\bar{x}$ = 43.9. However, had a lower number of psychosocial needs (r = -0.5) was negatively associated with MCS in women. For their part, men who presented better MCS, like their female counterparts, were: in work, $\bar{x}$ = 45.9; and had more than 7 friends, $\bar{x}$ = 42.7. In addition, unlike women, men living with someone presented better MCS, $\bar{x}$ = 39.0. However, had a lower number of psychosocial needs (r = -0.4) also was negatively associated with MCS in men (Table 4).

The variables associated with MCS that explained 38% of variance were: in work [$\beta_a$ = 6.1 (95% CI: 1.5;10.7)]; having a large number of friends [$\beta_a$ = 6.6(95% CI: 2.1;11.1)]; having a high number of psychosocial needs in the health area [$\beta_a$ = -3.9 (95% CI: -6;-1.8)]; and those perceiving that their social relationships and leisure activities had improved or not changed [$\beta_a$ = 8.6 (95% CI: 4.4;12.8)]. The last two were also associated with MCS in men in addition to living alone [$\beta_a$ = -7.1 (95% CI: -12.7;-1.4)]; lacking economic benefits [$\beta_a$ = 8.5 (95% CI: 3.2; 13.8)]; and having a high number of psychosocial [$\beta_a$ = -3.6 (95% CI: -7.1;-0.2)] and social area needs [$\beta_a$ = -7.7 (95% CI: -13.4;-2.0)]. These explain 39.2% of the variance in men. Clinical diagnosis was not associated with MCS in either men or women (Table 5).

## Discussion

The main results of this study are: 1) the mean PCS of HRQOL in women with SMI is significantly lower than that of men with SMI, whereas in the case of MCS, there are no observed statistically significant differences; 2) however, there are differences in the factors associated with PCS and MCS of HRQOL between women and men with SMI.

Consistent with other studies, we have found that women with SMI present a statistically lower PCS of HRQOL than men with SMI[15–17]. For example, it has been reported that women with panic disorders have lower mean scores of HRQOL in the physical activity scale than men[41]. These results highlight that there are gender differences in general in our society

**Table 2. Physical Component Score (PCS) of HRQOL by Gender.**

| | | WOMEN | | MEN | |
|---|---|---|---|---|---|
| | | M (IC95%) | sig. | M (IC95%) | sig. |
| **Civil Status** | Married | 42.8 (40:49.1) | 0.08 | 48.6 (45.1:52.1) | 0.82 |
| | No married (single, separated, widow) | 46.3 (43.5:49.1) | | 49.1 (46.5:51.7) | |
| **Education level** | Primary education | 41.8 (39.3:44.2) | 0.00 | 48.1 (44.7:51.5) | 0.17 |
| | Secondary education | 46.4 (42.8:50) | | 48.4 (45.5:51.4) | |
| | University studies | 51.7 (46.6:56.8) | | 54.4 (51.1:57.7) | |
| **Work Status** | Employed | 52.3 (49.5:55.1) | 0.00 | 52.2 (48.6:55.8) | 0.04 |
| | Unemployed (+students and paid sick leave) | 41.9 (39.6:44.1) | | 47.6 (45.1:50.1) | |
| **Living situation** | Alone | 46.9 (40.7:53.2) | 0.36 | 47.1 (40.8:53.3) | 0.37 |
| | With someone | 44.2 (42.1:46.3) | | 49.4 (47.3:51.6) | |
| **Number of friends** | 0–3 | 43.7 (40.6:46.8) | 0.66 | 48.8 (45.5:52.1) | 0.94 |
| | 4–6 | 44.4 (40.2:48.6) | | 49.5 (45.4:53.5) | |
| | ≥7 | 45.8 (42.6:49) | | 48.6 (45.3:52) | |
| **Legal status** | Not disabled | 45.6 (43.1:48.1) | 0.13 | 49.4 (47:51.7) | 0.48 |
| | Disabled | 42.4 (39.3:45.4) | | 47.6 (42.9:52.3) | |
| **Degree of impairment** | <33% | 46.2 (43.9:48.6) | 0.01 | 48.8 (46.4:51.2) | 0.84 |
| | >33% | 40.7 (37.2:44.3) | | 49.2 (45.2:53.3) | |
| **Degree of disability** | Yes | 41 (38.3:43.7) | 0.00 | 47.9 (44.2:51.6) | 0.42 |
| | No | 47.6 (44.9:50.3) | | 49.6 (47.2:52.1) | |
| **Economic benefits** | Receives benefits | 42.1 (39.8:44.4) | 0.00 | 47.9 (45.5:50.4) | 0.10 |
| | Does not receive benefits | 51.1 (48:54.2) | | 51.8 (48:55.6) | |
| **Monthly income** | <700 | 40.4 (37.4:43.4) | 0.00 | 47.5 (43.8:51.2) | 0.31 |
| | >700 | 47.1 (44.6:49.6) | | 49.7 (47.2:52.2) | |
| **Clinical diagnosis (Axis I and II)[a]** | Schizophrenia and other psychotic disorders | 45.3 (40.3:50.2) | 0.00 | 49.5 (45.9:53.2) | 0.35 |
| | Depression disorder | 40.5 (37.3:43.7) | | 45.6 (40.6:50.6) | |
| | Bipolar disorder I and II | 49.4 (46.4:52.5) | | 50.4 (47.1:53.6) | |
| | Others | 41.5 (36.7:46.3) | | 50.1 (45.4:54.8) | |
| **Comorbidity (Axis III)[a]** | No Comorbidity | 49.9 (46.5:53.2) | 0.00 | 51.6 (48.9:54.3) | 0.04 |
| | Comorbidity | 42.2 (39.9:44.5) | | 47.3 (44.5:50.2) | |
| **Social consequences and leisure activities** | Yes | 42.8 (40.5:45) | 0.00 | 47.6 (44.9:50.4) | 0.07 |
| | No | 50.2 (46.5:53.9) | | 51.6 (48.8:54.4) | |
| **Social relationships[b]** | Improved or no change | 49 (46.3:51.6) | 0.00 | 52.4 (49.8:55.1) | 0.01 |
| | Worse | 41.5 (38.7:44.2) | | 45.8 (42.6:49) | |
| | DK/NA | 43.9 (36:51.9) | | 47.2 (41.5:53) | |
| **Possibilities for the future** | Yes | 47.5 (44.7:50.3) | 0.01 | 50.1 (47.3:52.9) | 0.25 |
| | No | 42.1 (39.4:44.8) | | 47.7 (44.6:50.8) | |
| **Help in the future** | Yes | 41.5 (38.8:44.3) | 0.00 | 46.3 (43.1:49.5) | 0.01 |
| | No | 49.3 (45.1:53.5) | | 54.6 (51.6:57.5) | |
| | DK/NA | 47.4 (44.2:50.6) | | 50.1 (47.2:53) | |
| | | r (IC95%) | sig. | r (IC95%) | sig. |
| **Age** | | -0.3 (-0.2:-0.4) | 0.00 | -0.2 (-0.1:-0.3) | 0.06 |
| **Num. of children** | | -0.1 (-0.1:-0.2) | 0.10 | -0.2 (-0.1:-0.3) | 0.09 |
| **Num. of needs CAN[c]** | | -0.5 (-0.4:-0.6) | 0.00 | -0.5 (-0.4:-0.6) | 0.00 |
| **Type of needs[c]** | Basic | -0.2 (-0.1:-0.3) | 0.03 | - | - |
| | Social | -0.1 (0:-0.2) | 0.32 | -0.2 (-0.1:-0.3) | 0.03 |
| | Functioning | -0.3 (-0.2:-0.4) | 0.00 | -0.2 (-0.1:-0.3) | 0.04 |
| | Health | -0.6 (-0.5:-0.6) | 0.00 | -0.5 (-0.4:-0.6) | 0.00 |

*(Continued)*

**Table 2.** (Continued)

| | | WOMEN | | MEN | |
|---|---|---|---|---|---|
| | | **M (IC95%)** | **sig.** | **M (IC95%)** | **sig.** |
| | Services | -0.2 (-0.1:-0.3) | 0.04 | -0.2 (-0.1:-0.3) | 0.08 |

M: Mean; IC95%: Confidence Interval of 95%; r: Linear regression coefficient; DK: Does not know; NA: No answer

[a.] According to Manual DSM-IV

[b.] Social relationships and activities perceptions

[c.] According the Camberwell Assessment of Need Questionnaire (CAN-R)

and specifically, in people with mental health problems[42], both in psychiatric morbidity as well as in the pattern of behavior of different mental illnesses developed by men and women. The evidence indicates that the sociocultural factors that act through socially imposed roles and behavioral patterns are those that ultimately influence the way in which women and men display and cope with their psychological suffering. Over the last few decades, various studies have demonstrated that women often receive inferior and insensitive treatment by public institutions and by society in general, which either has no laws to prevent this behavior or fails to enforce them when it does[43].

However, unlike what has been reported in other studies[44], we did not find any significant statistical differences between women and men SMI with regard to MCS. For example, it has been reported that men with schizophrenia have worse MCS than women. This is explained by the fact that in the country under study, men have greater responsibility for supporting their families[44]. Nevertheless, this relationship was not observed in this study.

**Table 3. Model of the Physical Component Score of HRQOL by Gender.**

| WOMEN | | Coefficient (IC95%) | sig. | Adjusted coefficient (IC95%) | sig. |
|---|---|---|---|---|---|
| **Work Status** | Unemployed | 0 | | 0 | |
| | Employed | 10.4 (6.3:14.6) | 0.00 | 4.2 (0.3:8.2) | 0.03 |
| **Monthly income** | <700 | 0 | | 0 | |
| | >700 | 6.7 (2.8:10.6) | 0.00 | 4.4 (1:7.7) | 0.01 |
| **Comorbidity (Axis III)[a]** | Comorbidity | 0 | | 0 | |
| | No Comorbidity | 7.7 (3.6:11.7) | 0.00 | 4.5 (1.1:7.9) | 0.01 |
| **Age** | | -0.4 (-0.6:-0.2) | 0.00 | -0.2 (-0.4:0) | 0.01 |
| **Type of needs[b]** | Health | -6.5 (-8.2:-4.8) | 0.00 | -4.4 (-6.2:-2.6) | 0.00 |
| MEN | | Coefficient (IC95%) | sig. | Adjusted coefficient (IC95%) | sig. |
| **Education level** | Primary education | 0 | | 0 | |
| | Secondary | 0.3 (-4:4.6) | 0.89 | -3 (-6.6:0.6) | 0.11 |
| | University | 6.3 (-0.6:13.2) | 0.07 | 6.1 (0.4:11.7) | 0.04 |
| **Comorbidity (Axis III)[a]** | Comorbidity | 0 | | 0 | |
| | No Comorbidity | 4.3 (0.1:8.4) | 0.04 | 4.2 (0.7:7.7) | 0.02 |
| **Help in the future** | Yes | 0 | | 0 | |
| | No | 8.3 (3:13.5) | 0.00 | 4.6 (0.1:9.2) | 0.05 |
| **Type of needs[b]** | Health | -5.3 (-7.4:-3.3) | 0.00 | -5.6 (-7.7:-3.6) | 0.00 |
| | Services | -4.9 (-10.3:0.6) | 0.08 | -6.6 (-11.1:-2) | 0.01 |

IC95%: Confidence Interval of 95%

[a.] According to Manual DSM-IV

[b.] According the Camberwell Assessment of Need Questionnaire (CAN-R)

**Table 4. Mental Component Score (MCS) of HRQOL by gender.**

| | | WOMEN | | MEN | |
|---|---|---|---|---|---|
| | | M (IC95%) | sig. | M (IC95%) | sig. |
| **Civil Status** | Married | 35.9 (32.3:39.5) | 0.65 | 37 (31.8:42.3) | 0.81 |
| | No married (single, separated, widow) | 37 (33.8:40.2) | | 37.8 (34.3:41.2) | |
| **Education level** | Primary education | 32.7 (29.7:35.8) | 0.00 | 35.6 (31.6:39.6) | 0.12 |
| | Secondary education | 40.9 (36.7:45.1) | | 40.4 (36.1:44.6) | |
| | University studies | 40.6 (34.7:46.5) | | 32.2 (23.7:40.7) | |
| **Work Status** | Employed | 44.9 (40.7:49) | 0.00 | 45.9 (42:49.8) | 0.00 |
| | Unemployed (+students and paid sick leave) | 33.4 (30.8:36.1) | | 33.9 (30.5:37.2) | |
| **Living situation** | Alone | 37.5 (31.3:43.6) | 0.74 | 31.5 (26:37) | 0.03 |
| | With someone | 36.3 (33.7:38.9) | | 39 (35.8:42.2) | |
| **Number of friends** | 0–3 | 31.4 (27.8:35) | 0.00 | 33.2 (28.8:37.6) | 0.02 |
| | 4–6 | 37.4 (32.9:41.9) | | 36.1 (31.4:40.8) | |
| | ≥7 | 41.4 (37.5:45.3) | | 42.7 (37.7:47.6) | |
| **Legal status** | Not disabled | 37.8 (34.9:40.8) | 0.09 | 39 (35.7:42.3) | 0.05 |
| | Disabled | 33.4 (29.3:37.4) | | 32.5 (27.2:37.8) | |
| **Degree of impairment** | <33% | 36.4 (33.4:39.4) | 0.95 | 38 (34.4:41.6) | 0.66 |
| | >33% | 36.6 (32.6:40.5) | | 36.6 (31.9:41.4) | |
| **Degree of disability** | Yes | 32.3 (28.8:35.7) | 0.00 | 35.3 (30.9:39.6) | 0.22 |
| | No | 39.9 (36.8:43) | | 38.9 (35.1:42.7) | |
| **Economic benefits** | Receives benefits | 34 (31.2:36.7) | 0.00 | 34.1 (30.9:37.3) | 0.00 |
| | Does not receive benefits | 42.9 (38.8:47) | | 46.8 (42.5:51) | |
| **Monthly income** | <700 | 34.9 (30.9:38.9) | 0.33 | 36 (31.2:40.7) | 0.43 |
| | >700 | 37.3 (34.3:40.3) | | 38.3 (34.7:41.9) | |
| **Clinical diagnosis (Axis I and II)[a]** | Schizophrenia and other psychotic disorders | 38.9 (33.8:44) | 0.00 | 39.2 (34.3:44.2) | 0.45 |
| | Depression disorder | 33.1 (29.6:36.6) | | 36.2 (29.5:43) | |
| | Bipolar disorder I and II | 41.1 (37.2:44.9) | | 39.3 (34.3:44.4) | |
| | Others | 28.9 (20.4:37.5) | | 33.3 (27:39.5) | |
| **Comorbidity (Axis III)[a]** | No Comorbidity | 39.1 (34.4:43.8) | 0.14 | 35.8 (30.9:40.7) | 0.35 |
| | Comorbidity | 35.2 (32.5:38) | | 38.5 (35:42.1) | |
| **Social consequences and leisure activities** | Yes | 33.6 (31:36.3) | 0.00 | 34.3 (30.9:37.7) | 0.00 |
| | No | 45 (40.8:49.3) | | 43.9 (39.5:48.4) | |
| **Social relationships[b]** | Improved or no change | 43.9 (40.4:47.4) | 0.00 | 43.7 (39.9:47.4) | 0.00 |
| | Worse | 30.5 (27.7:33.3) | | 32.8 (28.9:36.8) | |
| | DK/NA | 40.7 (33:48.3) | | 30.1 (22:38.2) | |
| **Possibilities for the future** | Yes | 38.4 (34.8:42) | 0.13 | 37.1 (32.9:41.3) | 0.77 |
| | No | 34.8 (31.6:38) | | 38 (34.1:41.8) | |
| **Help in the future** | Yes | 31.9 (29:34.7) | 0.00 | 33.8 (30.3:37.4) | 0.00 |
| | No | 46.2 (41.1:51.3) | | 48 (42:54) | |
| | DK/NA | 38.1 (33.6:42.7) | | 37.2 (32.1:42.4) | |
| | | r (IC95%) | sig. | r (IC95%) | sig. |
| **Age** | | 0 (0:0.1) | 0.76 | -0.1 (0:-0.1) | 0.61 |
| **Num. of children** | | -0.1 (-0.1:-0.2) | 0.13 | 0 (0:0) | 0.97 |
| **Num. of needs CAN[c]** | | -0.5 (-0.4:-0.6) | 0.00 | -0.4 (-0.3:-0.5) | 0.00 |
| **Type of needs[c]** | Basic | -0.2 (-0.1:-0.3) | 0.02 | - | - |
| | Social | -0.2 (-0.2:-0.3) | 0.01 | -0.3 (-0.2:-0.4) | 0.01 |
| | Functioning | -0.3 (-0.3:-0.4) | 0.00 | -0.2 (-0.1:-0.3) | 0.11 |
| | Health | -0.5 (-0.4:-0.5) | 0.00 | -0.4 (-0.3:-0.5) | 0.00 |

*(Continued)*

**Table 4.** (Continued)

| | | WOMEN | | MEN | |
|---|---|---|---|---|---|
| | | **M (IC95%)** | **sig.** | **M (IC95%)** | **sig.** |
| | Services | -0.1 (-0.1:-0.2) | 0.20 | 0.1 (0:0.1) | 0.55 |

M: Mean; IC95%: Confidence Interval of 95%; r: Linear regression coefficient; DK: Does not know; NA: No answer

[a.] According to Manual DSM-IV

[b.] Social relationships and activities perceptions

[c.] According the Camberwell Assessment of Need Questionnaire (CAN-R)

In addition, we found in our study that some factors are associated with HRQOL in both women and men with SMI. As seen in other studies, having a high number of psychosocial needs in the area of health is associated with a worse PCS and MCS of HRQOL in both men and women[31–33]. Furthermore, not having physical comorbidity is associated with better PCS in women and in men, as has been reported in other studies[20,27,28].

On the other hand, we found some factors associated with HRQOL in persons with SMI in one of the two genders but not in the other. One of the factors that we found associated with HRQOL in women with SMI but not in men is age. This is generally negatively associated with HRQOL[18,19], especially in PCS[20]. This difference could be explained by women generally having a longer life expectancy but with more chronic illnesses and with fewer years of good health and, as a result, poorer quality of life than men[45]. These inequalities can be attributed to a more precarious labor market for women and therefore, a lower socioeconomic position with a double work load of paid employment and unpaid work in the home[46].

**Table 5. Model of the mental component score of HRQOL by gender.**

| WOMEN | | Coefficient (IC95%) | sig. | Adjusted coefficient (IC95%) | sig. |
|---|---|---|---|---|---|
| **Work Status** | Unemployed | 0 | | 0 | |
| | Employed | 11.4 (6.4:16.5) | 0.00 | 6.1 (1.5:10.7) | 0.01 |
| **Number of friends** | 0–3 | 0 | | 0 | |
| | 4–6 | 6 (0.3:11.7) | 0.04 | 4.2 (-0.6:8.9) | 0.08 |
| | ≥7 | 10 (4.6:15.4) | 0.00 | 6.6 (2.1:11.1) | 0.01 |
| **Social relationships[a]** | Worse | 0 | | 0 | |
| | Improved or no change | 13.5 (9:17.9) | 0.00 | 8.6 (4.4:12.8) | 0.00 |
| **Type of needs[b]** | Health | -6.5 (-8.7:-4.4) | 0.00 | -3.9 (-6:-1.8) | 0.00 |
| **MEN** | | Coefficient (IC95%) | sig. | Adjusted coefficient (IC95%) | sig. |
| **Living situation** | With someone | 0 | | 0 | |
| | Alone | -7.5 (-14.4:-0.6) | 0.03 | -7.1 (-12.7:-1.4) | 0.02 |
| **Economic benefits** | Receives | 0 | | 0 | |
| | Not receive benefits | 12.6 (6.8:18.5) | 0.00 | 8.5 (3.2:13.8) | 0.00 |
| **Social relationships[a]** | Worse | 0 | | 0 | |
| | Improved or no change | 10.8 (5.4:16.3) | 0.00 | 6.8 (1.4:12.1) | 0.01 |
| **CAN-R[b]** | Total | -3 (-4.7:-1.4) | 0.00 | -3.6 (-7.1:-0.2) | 0.04 |
| **Type of needs[b]** | Social | -5.2 (-9.1:-1.3) | 0.01 | -7.7 (-13.4:-2) | 0.01 |
| | Health | -5.6 (-8.6:-2.6) | 0.00 | -8.1 (-12.6:-3.6) | 0.00 |

IC95%: Confidence Interval of 95%

[a] Social relationships and activities perceptions

[b.] According the Camberwell Assessment of Need Questionnaire (CAN-R).

Level of education is not always associated with HRQOL[25,47,48], but in our study we observed that men with a higher level of education have better HRQOL than those that do not. Specifically, this association has been observed in PCS[18,20]. However, being in work is associated with better HRQOL in both components[21–23,49] for women. Both factors are indicators of socioeconomic position and according to the differences observed in our results we can say that they affect women and men differently, which leads us to believe that there are gender inequalities in HRQOL for people with SMI. Being in work seems to imply economic independence, which occurs later in women than in men, and makes them feel more integrated in society and able to contribute value through their abilities. Employment is thus a key tool for personal development and realization and is associated with better HRQOL. Nevertheless, gender inequalities continue to exist in employment (like wage inequality), the home (unequal work load for domestic tasks and their role as informal caregivers) and the broader community (continued acceptation of higher levels of violence against women)[50]. However, in men, this role is assumed to be traditional and, together with greater employability, results in this factor not explaining their HRQOL. On the other hand, level of education is the socioeconomic factor that is associated with better HRQOL in men. This fact can be explained because level of education ends up determining employment status, which influences the possibility and type of job position.

Other indicators of socioeconomic position are income and economic benefits, which are associated with better HRQOL[23,24] in both women and men. However, this association has been observed in different components according to gender, had income of greater than 700 euros/month is positively associated in PCS in women and lacking economic benefits are positively associated in MCS in men. These results agree with a report by the OECD on how increasing participation of women in economic life reduces gender inequality in the workplace and provides benefits for all[51]. In addition, it has been shown that social class, specifically belonging to a lower social class, is a key determinants of poor mental health in the general population, in both women and men[52].

A greater number of friends is also positively associated with MCS of HRQOL[21,22,31] in women. It is shown that social relations act as a protective factor for health[53] through different aspects including: access to direct help in the case of the need for care; a dissuasive effect that exercises social control over the practice of risky activities; a greater level of social integration that leads to access to resources; and the feeling of belonging to a group[54]. On the other hand, living alone is negatively associated with MCS of HRQOL[25] in men but not in women. Even though it was not observed in this study, there is a broad consensus that living together as a couple has a positive and immediate effect on the health of both men and women in the middle and long term, although in lesser intensity in the case of women[55,56]. These benefits are explained by three factors: reduction of risky and unhealthy habits[57]; creation and maintenance of a social support network to which they can turn to in critical situations [55]; and increased material well-being resulting from economies of scale by combining resources and the specialization of tasks between the couple[56].

In terms of social functioning, the total number and different areas of psychosocial needs is associated with HRQOL, as has been observed in numerous studies[22,31–33]. In this study, it was primarily seen in men. The psychosocial needs in the area of services is associated with PCS. This area of the CAN-R responds to aspects of information, telephone, transport and benefits related to a physical dimension. Psychosocial needs in the social area of the CAN-R, which refer to aspects of companionship, intimate relationships and sexual expression related to the psychological dimension, are associated with MCS. The total number of psychosocial needs are also associated with MCS of HRQOL because a greater number of psychosocial needs, and thus problems in daily life, in these persons worsens the MCS scores of HRQOL.

Current self-perception of improvement or maintenance of social relationships and leisure activities is associated with MCS in women and men. Self-perception is a belief, and believing that social relationships and leisure activities, and therefore the social network status (previously mentioned as a protective factor determinant of health), have been maintained or improved is associated with MCS of HRQOL[53,54].

## Strengths and limitations

SMI groups different clinical diagnoses together that share a series of characteristics. This is both a strength because there are not many studies of patients with these clinical characteristics. At the same time, it involves a certain difficulty when comparing results with other studies because most of them work with populations with a single diagnosis, mostly schizophrenia, or with different groupings of diagnosis that do not strictly correspond with the definition of SMI employed in this study.

There is a lot of variability among HRQOL studies of people with SMI, mostly because of different definitions of the concept (HRQOL) itself and at the same time from the variety of measurement instruments used. This study, however, used a validated instrument, SF-36. Furthermore, a lack of evidence has been observed concerning explicative variables of HRQOL in people with SMI according to gender.

## Conclusions

To improve HRQOL in people living in the community with SMI, priority should be given to programs or interventions that address the control and management of symptoms and clinical needs and socioeconomic aspects like level of education, reentering the labor force, economic level, etc., as well as improved development of social networks. However, these interventions also require a focus on gender to address the differences observed in the factors associated with HRQOL in women and men.

## Author Contributions

**Conceptualization:** Ester Colillas-Malet, Gemma Prat, Dolors Juvinyà.

**Data curation:** Ester Colillas-Malet, Albert Espelt.

**Formal analysis:** Ester Colillas-Malet, Albert Espelt.

**Investigation:** Ester Colillas-Malet, Albert Espelt.

**Methodology:** Ester Colillas-Malet, Gemma Prat, Albert Espelt, Dolors Juvinyà.

**Project administration:** Ester Colillas-Malet.

**Resources:** Ester Colillas-Malet, Gemma Prat.

**Software:** Ester Colillas-Malet, Albert Espelt.

**Supervision:** Gemma Prat, Dolors Juvinyà.

**Visualization:** Ester Colillas-Malet.

**Writing – original draft:** Ester Colillas-Malet.

**Writing – review & editing:** Gemma Prat, Albert Espelt, Dolors Juvinyà.

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
