## [Decision Letter · Decision Letter 0]

14 Oct 2019

PONE-D-19-20978

Gender Inequality in Health-Related Quality of Life in people with Severe Mental Illness

PLOS ONE

Dear Dra Colillas-Malet,

Thank you for submitting your manuscript to PLOS ONE. After careful consideration, we feel that it has merit but does not fully meet PLOS ONE’s publication criteria as it currently stands. Therefore, we invite you to submit a revised version of the manuscript that addresses the points raised during the review process.

We would appreciate receiving your revised manuscript by Nov 28 2019 11:59PM. To enhance the reproducibility of your results, we recommend that if applicable you deposit your laboratory protocols in protocols.io, where a protocol can be assigned its own identifier (DOI) such that it can be cited independently in the future. For instructions see: http://journals.plos.org/plosone/s/submission-guidelines#loc-laboratory-protocols

We look forward to receiving your revised manuscript.

Kind regards,

Geilson Lima Santana, M.D., Ph.D.

Academic Editor

PLOS ONE

Journal Requirements:

Reviewers' comments:

Reviewer's Responses to Questions

**Comments to the Author**

1. Is the manuscript technically sound, and do the data support the conclusions?

Reviewer #1: Partly

Reviewer #2: Yes

2. Has the statistical analysis been performed appropriately and rigorously? 

Reviewer #1: I Don't Know

Reviewer #2: Yes

3. Have the authors made all data underlying the findings in their manuscript fully available?

Reviewer #1: No

Reviewer #2: Yes

4. Is the manuscript presented in an intelligible fashion and written in standard English?

Reviewer #1: Yes

Reviewer #2: Yes

5. Review Comments to the Author

Reviewer #1: The current study explored the gender differences of HRQOL among individuals with severe mental illness in Spain. The study is interesting and the authors presented well. However, I do feel more information is still needed for me to really judge whether the analyses were done properly. The following please find my comments for authors' reference

1. Please add in more details for the study process including simple random sampling method used in the current study, like how the sample frame was determined, who approached the patients at the first place, how to avoid potential coercion, any inconvenience fees, and etc. The current description is very vague.

2. Were the data collected through interview or self-administered questionnaire? In the sample selection authors mentioned through interview; however in the variables authors then mentioned 'it's a self-administer questionnaire...' This is confusing, please make it clear.

3. For the CAN-R - 1) please spell in full in when it first appear in the manuscript, and put the abbreviation in the brackets; 2) is the tool validated among people with severe metal illness in Spain? also how about the domains? In the citing reference, their study sample comprised individuals with schizophrenia from 5 different European countries, which is quite different from the current study sample?

4. Variables such as 'social consequences', 'possibilities for the future', 'help in the future' and 'economic benefits' were very confusing, please give more details.

5. How was 'num. of need CAN' determined? This is unclear throughout the whole paper.

6. For work status - shouldn't people with paid sick leave also fall under employed?

7. Why was 'number of friends' not a continuous variable?

8. For social relationships - compared to when?

9. I'm not quite sure how the authors conducted the regression analyses. In the current paper the description was not clear. I need more details to tell whether the statistical analyses were done properly.

10. I need the authors to clarify the above mentioned confusions before I can make a proper judgement on the discussion.

Reviewer #2: PONE-D-19-20978

Gender Inequality in Health-Related Quality of Life in people with Severe Mental Illness

PLOS ONE

This study is interesting because it demonstrates the effects that chronic mental illnesses leave on patients for the rest of their lives.

The article is well done, I congratulate the authors. I only have a few very punctual questions to consider and improve the manuscript.

MINOR REVISION

1. In line 21, directly writing the acronym “SMI”, it is important to explain what this means.

2. In paragraph 29 through 31, the sentence “There are many people with mental health problems in the community and their HRQOL is especially important to consider. They often live in more disadvantaged conditions compared to the rest of the general population.” You have to put the references and explain why this is what you describe.

3. The same, in the sentence of lines 31-32 "They often live in more disadvantaged conditions compared to the rest of the general population". Must have references.

4. At line 79, “Spanish version [32] of the Medical Outcome Study Short Form 36 (SF-36)” I suggest writing the name of the questionnaire in Spanish, too.

5. Similarly, line 94, written “the Spanish version of the CAN-R [34]”, has to describe what it means as well as in English and Spanish.

Best regards.

6. PLOS authors have the option to publish the peer review history of their article (what does this mean?). If published, this will include your full peer review and any attached files.

Reviewer #1: No

Reviewer #2: No

---

## [Author Response · Author response to Decision Letter 0]

26 Nov 2019

Dear Editor,

In first place we want to thank you for the new reviewers’ comments. Below you can find the answers to each of the comments and all changes made are highlighted in track changes mode activated in the manuscript:

REVIEWER 1

COMMENT 1: 

Please add in more details for the study process including simple random sampling method used in the current study, like how the sample frame was determined, who approached the patients at the first place, how to avoid potential coercion, any inconvenience fees, etc. The current description is very vague.

AUTHORS ANSWER: It has been rewritten the sample selection section adding more details for the study process: There was a total of 783 people who met the inclusion criteria (see line 63). The sample was selected by simple random sampling and the size was determined based on the known size of the population that met inclusion criteria, with a 5% error and a 3% accuracy. The theoretical sample was 258 people diagnosed with SMI who carried out ambulatory follow-up at mental health center. (see line 70-73)

COMMENT 2:

Were the data collected through interview or self-administered questionnaire? In the sample selection authors mentioned through interview; however, in the variables authors then mentioned 'it's a self-administer questionnaire...' This is confusing, please make it clear.

AUTHORS ANSWER: It has been deleted the word “self-administred”, see line 83, because the questionnaire finally was administered by a trained interviewer. 

COMMENT 3:

For the CAN-R - 1) please spell in full in when it first appears in the manuscript, and put the abbreviation in the brackets; 2) is the tool validated among people with severe mental illness in Spain? also how about the domains? In the citing reference, their study sample comprised individuals with schizophrenia from 5 different European countries, which is quite different from the current study sample?

AUTHORS ANSWER: 1)It has been added the complete name of the CAN-R, Camberwell Assessment of Need, see line 97; 2) The tool has been used is validated among people with schizophrenia, considered a long-acting psychotic disorder such as severe mental illness; The 22 domains can also be grouped in five dimensions (CAN manual, Research version 3.0-E): 

A. Basic (3 domains): accommodation, food and daytime activities 

B. Health (7 domains): physical health, psychotic symptoms, psychological distress, safety to self, safety to others, alcohol and drugs

C. Social (3 domains): company, intimate relationships and sexual expression

D. Functioning (5 domains): looking after the home, self-care, childcare, education and money

E. Services (4 domains): information, telephone, transport and benefits.

The CAN-EU has good inter-rater and test-retest reliability, the authors has been added a new reference, number 39 (see line 106).

COMMENT 4:

Variables such as 'social consequences', 'possibilities for the future', 'help in the future' and 'economic benefits' were very confusing, please give more details.

AUTHORS ANSWER: It has been explained more details about these variables: as for example, if the SMI had had consequences on their social life and leisure activities, if these social consequences had improved or worsened their social relationships, if they had considered what possibilities would be in the future and if they thought that would need help. (see lines 107-110) 

COMMENT 5:

How was 'num. of need CAN' determined? This is unclear throughout the whole paper.

AUTHORS ANSWER: It has been modified how was number of needs CAN determined: All the areas are scored from 0 to 1 with: 0, no problem; 1, moderate or serious problem due to help given. (see lines 103-104)

COMMENT 6: 

For work status - shouldn't people with paid sick leave also fall under employed?

AUTHORS ANSWER: The authors divided this variable in two categories (employed/unemployed) because we considered if the participant was working in the moment of the interview or not. 

COMMENT 7: 

Why was 'number of friends' not a continuous variable?

AUTHORS ANSWER: This is a categorical variable because from the beginning it was posed as such. In any case, this consideration will be taken in account for upcoming investigations.

COMMENT 8: 

For social relationships - compared to when?

AUTHORS ANSWER: It has been explained more details about this variable: as for example, if the SMI had had consequences on their social life and leisure activities, if these social consequences had improved or worsened their social relationships, if they had considered what possibilities would be in the future and if they thought that would need help. (see line 109)

COMMENT 9:

I'm not quite sure how the authors conducted the regression analyses. In the current paper the description was not clear. I need more details to tell whether the statistical analyses were done properly.

AUTHORS ANSWER: The authors has been explained more details of the statistical analyses: The variables that presented a p<0.1 in the bivariate model were included in the multivariate models. In addition, the colineality of these variables was studied. (see lines 121-123)

REVIEWER 2

COMMENT 1:

In line 21, directly writing the acronym “SMI”, it is important to explain what this means.

AUTHORS ANSWER: It has been added the abbreviation in line 4, that is in the first place that this concept appears.

COMMENT 2: 

In paragraph 29 through 31, the sentence “There are many people with mental health problems in the community and their HRQOL is especially important to consider. They often live in more disadvantaged conditions compared to the rest of the general population.” You have to put the references and explain why this is what you describe.

AUTHORS ANSWER: It has been added references in lines 29-31.

COMMENT 3: 

The same, in the sentence of lines 31-32 "They often live in more disadvantaged conditions compared to the rest of the general population". Must have references.

AUTHORS ANSWER: It has been added references in lines 31-32.

COMMENT 4: 

At line 79, “Spanish version [32] of the Medical Outcome Study Short Form 36 (SF-36)” I suggest writing the name of the questionnaire in Spanish, too.

AUTHORS ANSWER: It has been written the name of the questionnaire in Spanish: Cuestionario de Salud SF-36. (see line 83)

COMMENT 5:

Similarly, line 94, written “the Spanish version of the CAN-R [34]”, has to describe what it means as well as in English and Spanish.

AUTHORS ANSWER: It has been written the name of the questionnaire in Spanish: Cuestionario Camberwell para la evaluación de necesidades. (see line 98)

We hope that the changes done in the manuscript and the responses given to the comments are of your compliance, making possible the acceptance of the manuscript for being published on PlosOne.

We thank you in advance your evaluation. 

Sincerely,

Authors.

---

## [Decision Letter · Decision Letter 1]

11 Dec 2019

PONE-D-19-20978R1

Gender Inequality in Health-Related Quality of Life in people with Severe Mental Illness

PLOS ONE

Dear Dra Colillas-Malet,

Thank you for submitting your manuscript to PLOS ONE. After careful consideration, we feel that it has merit but does not fully meet PLOS ONE’s publication criteria as it currently stands. Therefore, we invite you to submit a revised version of the manuscript that addresses the points raised during the review process.

We would appreciate receiving your revised manuscript by Jan 25 2020 11:59PM. To enhance the reproducibility of your results, we recommend that if applicable you deposit your laboratory protocols in protocols.io, where a protocol can be assigned its own identifier (DOI) such that it can be cited independently in the future. For instructions see: http://journals.plos.org/plosone/s/submission-guidelines#loc-laboratory-protocols

We look forward to receiving your revised manuscript.

Kind regards,

Geilson Lima Santana, M.D., Ph.D.

Academic Editor

PLOS ONE

Reviewers' comments:

Reviewer's Responses to Questions

**Comments to the Author**

1. If the authors have adequately addressed your comments raised in a previous round of review and you feel that this manuscript is now acceptable for publication, you may indicate that here to bypass the “Comments to the Author” section, enter your conflict of interest statement in the “Confidential to Editor” section, and submit your "Accept" recommendation.

Reviewer #1: (No Response)

2. Is the manuscript technically sound, and do the data support the conclusions?

Reviewer #1: Partly

3. Has the statistical analysis been performed appropriately and rigorously? 

Reviewer #1: Yes

4. Have the authors made all data underlying the findings in their manuscript fully available?

Reviewer #1: No

5. Is the manuscript presented in an intelligible fashion and written in standard English?

Reviewer #1: Yes

6. Review Comments to the Author

Reviewer #1: I would like to thank the authors for providing me more details of the data analysis part, which makes the paper clearer. However, after reviewing the responses and the revised manuscript, I have some further comments need to seek authors clarifications.

Regarding authors' responses, my comment 2 - author suggested that they had clarified self-administered in line 83, which I didn't see, please clarify.

my comment 6 - I'm not convinced by the current explanation. in our local context, normally individuals would only take 1 day sick leave (paid), unless they had severe conditions and have to be hospitalized. In this case, it makes more sense to be categorized into employed group. Please provide more specified reason why paid sick leave should be categorized into unemployed group if there are any cultural differences (and add the details into the revised manuscript).

Regarding the revised manuscript (all refers to the line numbers in manuscript with track changes)

1) the way authors presenting the regression results is confusing - please use different ways for associations with different directions. Take PCS among women for example, better PCS is POSITIVELY associated with age, but NEGATIVELY associated higher number of health related needed.

2) also for the results on regressions - please list out the reference group if there are three or more categories for a variable (for 2 categories, it is also recommended to list out the reference group, it's more clear).

3) In discussion, line 205-106 - according to the regression results, i thought co-morbidity is only significant among PCS, not among all HRQOL, right? Pls don't over interpret your findings.

4) in discussion line 210-212 - Are you sure life expectancy is a good explanation? according to your descriptive statistics, I don't think the age range is huge that can cause such gender differences.

5) In discussion line 233-237 - for income, isn't it only significant among MCS women? Where are these 'PCS in women' and 'MCS in men' come from?

6) after reading the whole article, I didn't see any gender inequity evidences other than men have better PCS scores, and this was only from a non-adjusted comparison. All other findings were about gender differences. In this case I think the manuscript title should be revised to 'gender differences ...'

7. PLOS authors have the option to publish the peer review history of their article (what does this mean?). If published, this will include your full peer review and any attached files.

Reviewer #1: No

---

## [Author Response · Author response to Decision Letter 1]

20 Jan 2020

Dear Editor,

In first place we want to thank you for the new reviewer’ comments. Below you can find the answers to each of the comments and all changes made are highlighted in track changes mode activated in the manuscript:

REVIEWER

COMMENT 1: 

Regarding authors' responses, my comment 2 - author suggested that they had clarified self-administered in line 83, which I didn't see, please clarify.

AUTHORS ANSWER: It has been rewritten the sample selection section adding more details for the study process: The participants were recruited by telephone, which also served to specify the date and time for the interview. The interview consisted of a 30-minute evaluation where a trained interviewer filled out the various questionnaires. (see lines 77-79)

COMMENT 2:

My comment 6 - I'm not convinced by the current explanation. In our local context, normally individuals would only take 1 day sick leave (paid), unless they had severe conditions and have to be hospitalized. In this case, it makes more sense to be categorized into employed group. Please provide more specified reason why paid sick leave should be categorized into unemployed group if there are any cultural differences (and add the details into the revised manuscript).

AUTHORS ANSWER: The authors categorized this variable thus because, as argued in the discussion section: being in work makes women feel more integrated in society and able to contribute value through their abilities. Employment is thus a key tool for personal development and realization and is associated with better HRQOL. Therefore, beyond being an indicator of socioeconomic position also is an indicator for personal development and realization and this aspect was of particular interest to us we had other variables that reflect aspects of socioeconomic position. 

COMMENT 3: 

The way authors presenting the regression results is confusing - please use different ways for associations with different directions. Take PCS among women for example, better PCS is POSITIVELY associated with age, but NEGATIVELY associated higher number of health related needed.

AUTHORS ANSWER: It has been rewritten this part in the paragraph of PCS: The PCS for HRQOL shows lower scores in women with SMI than men with SMI (x ®women=44.6 and x ®men=49.0; p=0.00) (Table 1). The women presenting better PCS were: unmarried, x ®=46.3; had higher levels of education/college educated, x ®=51.7; in work, x ®=52.3; had bipolar disorder, x ®=49.4; without physical comorbidity, x ®=49.9; and had seen their social relationships and leisure activities as having improved or not changed, x ®=49.0. However, age (r=-0.3) and had a lower number of psychosocial needs (r=-0.5) was negatively associated with PCS in women. On the other hand, men, like women, who presented better PCS were: in work, x ®=52.2; without physical comorbidity, x ®=51.6; and had seen their social relationships and leisure activities as having improved or not changed, x ®=52.4. However, age (r=-0.2) and had a lower number of psychosocial needs (r=-0.5) also was negatively associated with PCS in men. (see lines 141-150)

Also, it has been rewritten this part in the paragraph of MCS: Men and women had similar MCS scores for HRQOL (x ®women=36.4 and x ®men=37.5) (Table 1). The women who had better MCS were those who: had secondary education, x ®=40.9; were in work, x ®=44.9; had more than 7 friends, x ®=41.4; had bipolar disorder, x ®=41.1; and those that perceived that their social relationships and leisure activities had improved or not changed, x ®=43.9. However, had a lower number of psychosocial needs (r=-0.5) was negatively associated with MCS in women. For their part, men who presented better MCS, like their female counterparts, were: in work, x ®=45.9; and had more than 7 friends, x ®=42.7. In addition, unlike women, men living with someone presented better MCS, x ®=39.0. However, had a lower number of psychosocial needs (r=-0.4) also was negatively associated with MCS in men. (see lines 162-170)

COMMENT 4: 

Results on regressions - please list out the reference group if there are three or more categories for a variable (for 2 categories, it is also recommended to list out the reference group, it's more clear).

AUTHORS ANSWER: It have added the reference categories to the categorical variables in tables 3 and 5. 

COMMENT 5:

In discussion, line 205-106 - according to the regression results, i thought co-morbidity is only significant among PCS, not among all HRQOL, right? Pls don't over interpret your findings.

AUTHORS ANSWER: It has been rewritten this sentence because the comorbidity has been only significant among PCS: Furthermore, not having physical comorbidity is associated with better PCS in women and in men, as has been reported in other studies. (see lines 206-207).

COMMENT 6:

In discussion line 210-212 - Are you sure life expectancy is a good explanation? according to your descriptive statistics, I don't think the age range is huge that can cause such gender differences.

AUTHORS ANSWER: In the table 1 of the result it can be observed that women have a significantly higher mean age than men (x ®women=44.9 years and x ®men=41.3 years; p=0.01), and this factor together with the double work load, we considered can be explain this differences by gender. 

COMMENT 7: 

In discussion line 233-237 - for income, isn't it only significant among MCS women? Where are these 'PCS in women' and 'MCS in men' come from?

AUTHORS ANSWER: The authors are revised and rewritten this paragraph: Other indicators of socioeconomic position are income and economic benefits, which are associated with better HRQOL[23,24] in both women and men. However, this association has been observed in different components according to gender, had income of greater than 700 euros/month is positively associated in PCS in women and lacking economic benefits are positively associated in MCS in men. These results agree with a report by the OECD on how increasing participation of women in economic life reduces gender inequality in the workplace and provides benefits for all[51]. In addition, it has been shown that social class, specifically belonging to a lower social class, is a key determinants of poor mental health in the general population, in both women and men[52]. (see lines 234-242) 

COMMENT 8: 

After reading the whole article, I didn't see any gender inequity evidences other than men have better PCS scores, and this was only from a non-adjusted comparison. All other findings were about gender differences. In this case I think the manuscript title should be revised to 'gender differences ...'

AUTHORS ANSWER: The authors are revised and rewritten the title: Gender Differences in Health-Related Quality of Life in people with Severe Mental Illness. (see lines 1-2)

We hope that the changes done in the manuscript and the responses given to the comments are of your compliance, making possible the acceptance of the manuscript for being published on PlosOne.

We thank you in advance your evaluation. 

Sincerely,

Authors.

---

## [Editor Report · Decision Letter 2]

3 Feb 2020

Gender Diferences in Health-Related Quality of Life in people with Severe Mental Illness

PONE-D-19-20978R2

Dear Dr. Colillas-Malet,

We are pleased to inform you that your manuscript has been judged scientifically suitable for publication and will be formally accepted for publication once it complies with all outstanding technical requirements.

With kind regards,

Geilson Lima Santana, M.D., Ph.D.

Academic Editor

PLOS ONE

---

## [Editor Report · Acceptance letter]

12 Feb 2020

PONE-D-19-20978R2 

Gender Differences in Health-Related Quality of Life in people with Severe Mental Illness 

Dear Dr. Colillas-Malet:

I am pleased to inform you that your manuscript has been deemed suitable for publication in PLOS ONE. Congratulations! Your manuscript is now with our production department. 

With kind regards,

on behalf of

Dr. Geilson Lima Santana 

Academic Editor

PLOS ONE